# Magnetism and Grain-Size Distribution of Particles Deposited on the Surface of Urban Trees in Lanzhou City, Northwestern China

**DOI:** 10.3390/ijerph182211964

**Published:** 2021-11-14

**Authors:** Bo Wang, Xiaochen Zhang, Chenming Gu, Mei Zhang, Yuanhao Zhao, Jia Jia

**Affiliations:** College of Geography and Environmental Sciences, Zhejiang Normal University, Jinhua 321004, China; zhangxiaochen@zjnu.edu.cn (X.Z.); chenmgu@zjnu.edu.cn (C.G.); zhangmei@zjnu.edu.cn (M.Z.); zhaoyuanhao@zjnu.edu.cn (Y.Z.); jiaj@zjnu.edu.cn (J.J.)

**Keywords:** PM pollution, urban forest, biomagnetic monitoring, grain-size distribution, environmental conditions

## Abstract

Studies on the variation in the particulate matter (PM) content, Saturation Isothermal Remanent Magnetization (SIRM), and particle grain-size distribution at a high spatial resolution are helpful in evaluating the important role of urban forests in PM removal. In this study, the trees located in dense urban forests (T0) retained more PM than trees located in open spaces (T1–T4); the SIRM and PM weight of T0 were 1.54–2.53 and 1.04–1.47 times more than those of T1–T4, respectively. In addition, the SIRM and PM weight decreased with increasing distance to the road, suggesting that distance from pollution sources plays a key role in reducing the air concentration of PM. The different grain-size components were determined from frequency curve plots using a laser particle-size analyzer. A unimodal spectrum with a major peak of approximately 20 μm and a minor peak between 0.1 and 1 μm was observed, indicating that a large proportion of fine air PM was retained by the needles of the study trees. Additionally, more <2.5 μm size fraction particles were observed at the sampling site near the traffic source but, compared to a tree in a row of trees, the percentage of the >10 μm size fraction for the tree in the dense urban forest was higher, indicating that the particles deposited on the needle surface originating from traffic sources were finer than those from natural atmospheric dust. The exploration of the variation in the PM weight, SIRM, and grain size of the particles deposited on the needle surface facilitates monitoring the removal of PM by urban forests under different environmental conditions (e.g., in closed dense urban forests and in open roadside spaces), different distances to roads, and different sampling heights above the ground.

## 1. Introduction

Atmospheric particulate matter (PM), especially fine PM, has an adverse impact on human health [1,2,3,4,5,6]. These studies found that the main sources of ambient PM_2.5_ globally are traffic (25%), industry (15%), domestic fuel burning (20%), unspecified sources of human origin (22%), and natural dust and salt (18%) based on 419 source apportionment records, and traffic has been targeted as an important contributor to ambient air pollution in urban areas. Finer particles are more toxic than larger particles with the same chemical composition and have adverse health effects [2,3,4] due to their ability to penetrate deep into the respiratory tract and lungs and then induce respiratory and lung disease, immune system disorders, and even premature death [7].

The role of urban forests in PM removal has been verified both in experiments [8,9,10,11,12] and model simulations [13,14,15]. For instance, approximately 234 tons of PM_10_ were removed by Chicago’s trees in 1991 [16], 772 tons of PM_10_ were removed in Beijing [17], and Santiago’s urban forests were effective in removing PM_10_ [18]. Selmi et al. [19] found with an integrated i-Tree Eco model estimate that public trees removed approximately 12 tons of PM_2.5–10_ and 5 tons PM_2_ in 2012 [20]. However, urban forests have different PM removal capacities due to their composition of different tree species [21,22,23,24], sampling heights [22,25], tree configurations [26], ventilation conditions/tree locations [25], leaf traits [27,28,29], and exposure times [30]. For instance, coniferous species always have higher dust removal capacities than broadleaf species [31,32,33] due to their longer exposure time in the polluted environment and complex leaf structure. Previous studies have shown that the evergreen coniferous *Juniperus formosana* is an effective remover of air PM in arid and semiarid areas due to its dense canopy and extensive plantings [26,34].

Biomagnetic monitoring is regarded as an effective tool [13,35,36,37,38,39,40,41] to evaluate the capacity of urban plants to remove particulates, and Saturation Isothermal Remanent Magnetization (SIRM) is one of the most commonly used magnetic parameters. The distance and intensity of pollution sources are important factors affecting leaf magnetism; usually, higher leaf SIRM values are observed in high traffic areas [35,42] and industrial areas [43,44] than in parks or residential areas as well as in soils [45,46,47,48], sediments [49], street dust [50], and falling dust [51,52].

The grain size of particles deposited on the leaf surface showed significant differences between different species (e.g., the coniferous *Pinus mugo* showed the largest accumulation of the PM_0.2_ fraction) [22] and functional areas [9]. Larger particle size fractions (sum of >10 μm, 3–10 μm, and 0.2–3 μm size fractions) were found at the bottoms of tree crowns than in the middle and top areas [53]. Scanning electron microscopy (SEM) [9,40] and filters [22,53,54] are generally used to measure the concentration of different grain sized particles on the leaf surface. In this study, a Master size 2000 laser particle-size analyzer was used to measure the percentage of particle volume weighted means because of the advantages of a wide measurement range, high precision, fast speed of data collection, and high precision of small particle measurement associated with this method.

This study reports the variation in the PM weight, SIRM, and grain size of particles deposited on the surface of *Juniperus formosana* needles and the possible influencing factors: for instance, locational, distance to the road, and sampling height (if any). Thus, the aims of the present study are as follows: (i) to explore the differences in PM weight, SIRM, and grain-size distribution between an enclosed urban forest and open road environment and (ii) to evaluate the variation in PM weight, SIRM, and grain-size distribution with increasing distance to the road and sampling height above the ground.

## 2. Materials and Methods

### 2.1. Sampling and Experiment

Lanzhou City is situated in the northwest of China. The topography of Lanzhou City is a basin surrounded by mountains, and the Yellow River flows through the center of the city. Five trees (T0–T4) were selected in Lanzhou city to examine the differences in particle accumulation in different environments. Sampling site T0 was located on the campus of Lanzhou University (103°51′29.87′′ N, 36°2′52.96′′ E) (Figure 1), which has many green trees and can be regarded as an enclosed dense urban forest. Sampling sites T1, T2, T3, and T4 were located along Beibinhe Road (36°05′25.39″ N, 103°43′58.79″ E) in the Anning district in an open road space. T1 and T2 were both approximately 6 m away from the road, and T3 and T4 were 10 m and 24 m away from the road, respectively. T1 was located in a row of trees, and T2, T3, and T4 were solitary trees that were unaffected by other trees in the two directions parallel to the road axis. T0–T4 were approximately 5–6 m high, and all samples were collected from 120 to 600 cm (T0), 100 to 540 cm (T1), and 100 cm to 560 cm (T2, T3, and T4), respectively, in April 2017 at approximately 20 cm intervals. The perimeter of the trees is ≈4 m. Fully developed undamaged leaves were collected from the outer canopy of each tree. In addition, to avoid possible magnetic differences caused by the maturity of the needles, only mature leaves were collected from the outer canopy.

All mature *Juniperus formosana* needles were collected before budbreak from four sites in four directions (NE, SW, NW, and SE). The isothermal remanent magnetization (IRM) acquired in the maximum field of 1 T is defined as the saturation isothermal remanent magnetization (SIRM, 10^−5^ Am^2^ kg^−1^) and reflects the contribution of ferrimagnetic and antiferromagnetic minerals; it is a common indicator of the concentration of magnetic particulate matter in biomonitoring studies. SIRM was performed with an MMPM10 pulsed magnetizer and remanence was measured after the application of a field of 1 T. Then, the samples were measured with a JR-6A spinner magnetometer. The magnetism of the coniferous needles and the weight of the PM deposited over the needles were detected according to the protocol described in Chen et al. [26]. The SIRM of the particles deposited on the needles was calculated by the difference before and after washing the needles, and the weight of the particles deposited on the needles was detected and analyzed following the elution–filtration method.

Then, the grain-size distribution was measured. A brief description of the grain-size distribution procedure is as follows: Duplicate needles were washed with 150 mL of distilled water, brushed carefully, and then placed in an ultrasonicator (KQ-500DE) for 20 min to ensure that the PM was completely removed from the surface. The washings were passed through a nylon sieve (mesh diameter 100 μm) to remove impurities. Then, the samples were dispersed in 10 mL of (NaPO_3_)_6_ solution. The sample solution was ultrasonicated for 10 min. The grain-size distribution was measured using a Master size 2000 laser particle-size analyzer (detection range: 0.02–2000 μm).

All of the above analyses were carried out in the Key Laboratory of Western China’s Environmental Systems (Ministry of Education), Lanzhou University.

### 2.2. Statistical Analysis

Data analysis was performed using SPSS 22.0 (IBM, Solutions Statistical Package for the Social Sciences) and Origin 2018 (OriginLab). Significant differences in the PM weight, SIRM, and grain size of particles between different sampling sites, sampling directions, and height ranges were tested using one-way ANOVA. The normality of the data was evaluated statistically using the Kolmogorov–Smirnov test, and then, the results were logarithm transformed to meet the normality assumptions.

## 3. Results

### 3.1. The Grain-Size Characteristics of the Particles

The grain-size frequency of distribution and cumulative distribution provide reference information about the particle distribution types [55]. The grain-size frequency of distribution of the particles deposited on the surfaces of all leaves generally showed a normal distribution and could be recognized as a unimodal spectrum on a frequency curve, generally ranging from 0.1 to 100 μm, characterized by a dominant fraction with a major peak at approximately 20 μm (Figure 2a). In addition, a minor peak between 0.1 and 1 μm was observed (Figure 2a), suggesting that a large proportion of fine particles were deposited over the surface of *Juniperus formosana* needles. The particles were dominated by the >10 μm and the 2.5–10 μm size fractions (62.54% and 27.17%, respectively; Figure 2b), followed by the 1–2.5 μm and 0.02–1 μm size fractions (6.09% and 4.19%, respectively; Figure 2b). The <10 μm size fraction accounted for 37.46% of all particles, which is consistent with the previous results reported by Wang et al. [56], who confirmed that the <10 μm size fraction of the particles deposited on *Cedrus deodara* needles accounted for 33.97% of those particles in the central urban area of Nanjing, China, where air pollution is heavy.

The SIRM, PM weight, and grain-size distribution in the different size fractions (0.02–1 μm, 1–2.5 μm, 2.5–10 μm, >10 μm) of particles deposited on the surface of needles at sites T0–T4 are shown in Appendix A. The SIRM values ranged from 2.08 × 10^−5^ to 659.49 × 10^−5^ Am^2^·kg^−1^, and the concentration of the PM ranged from 0.56 to 67.69 g·kg^−1^. The SIRM and PM weight at site T0, located on campus, were substantially higher than those of T1–T4, located in an open roadside space (Appendix A). The SIRM of T1, which was located in a row of trees, was higher than that of T2, which was a solitary tree, even though these sites were both 6 m away from the road. Simultaneously, decreasing SIRM and weight were observed with increasing distance from the road (T2 > T3 > T4). The 0.02–1 μm and 1–2.5 μm size fractions (4.56%, and 6.61%, respectively) were most abundant in T1 but least abundant in T0 (3.67%, and 5.47%, respectively), and this size fraction was more abundant in T2 than T3 and T4 (Appendix A). The 2.5–10 μm size fractions of T0 and T1 were always more abundant than those of T2, T3, and T4. Conversely, the >10 μm size fractions of T0 (63.27%), T2 (63.22%), T3 (63.81%), and T4 (62.33%) were more abundant than that of T1 (59.93%).

### 3.2. Variations in Grain-Size Fraction

To determine whether there were statistically significant differences in the SIRM, the weight of PM, and different grain size fractions (0.02–1 μm, 1–2.5 μm, 2.5–10 μm, and >10 μm), one-way ANOVA was performed on the results from all sites, and significant differences (*p* < 0.05) were observed for the results of T1 and T2 and the 2.5–10 μm size fraction of T1 between the four sampling directions (Table 1 and Appendix A). However, nonsignificant differences were found between the four sampling directions at sites T3 and T4, suggesting that dust retention by the trees near the road (T1 and T2) differs from that of those trees that are far away from the road and located in the enclosed dense urban forest.

### 3.3. Variations among Different Locations

As shown in Figure 3, the SIRM and the weight of PM deposited on the needles at site T0 in the SW and SE directions were always higher than those at sites T1–T4; the values of T0 were lower than those of T1 and T2 in the NE direction, while the values of T1 were higher than those of T2 in the NE direction, but the opposite results were true in the other directions (SW, NW, and SE). Additionally, the SIRM and weight decreased with increasing (from T2 to T4) distance in each sampling direction, except in the SW direction. Slight reductions in the PM weight and SIRM (19.81–26.09% and 22.23–28.57%, respectively) were observed in the NE, NW, and SE directions at site T3, where the sampling distance increased to 10 m from the road (Figure 3b). However, greater reductions were observed at site T4. In the NE direction, the PM weight and SIRM of T4 were 47.01% and 64.10% less than that of T2 (Figure 3a). The values were 35.49% and 34.10% in the NW direction (Figure 3c) and 27.20% and 39.53% in the SE direction (Figure 3d). The results showed that the trees in the enclosed dense urban forest accumulated more airborne particles with their canopy, and with increasing distance from the road, the weight of PM deposited on the leaves decreased.

The percentages of the 0.02–1 μm and 1–2.5 μm size fractions decreased in the sequence of T1/T2 > T3/T4 > T0 (Figure 4a,b); the greatest concentration of 2.5–10 μm size fraction particles was observed at T1 (Figure 4c). In contrast, the lowest concentration of the >10 μm size fraction was observed at T1 (Figure 4d). The results showed that more fine particles were deposited on the surface of needles on the trees close to the road.

### 3.4. Variations among Different Sampling Heights

Previous work showed that PM retention on the tree canopy is complex and affected by sampling height. The hierarchical clustering method was used to test the impact of tree height on the deposition of PM, and the final cluster members were divided into three height ranges (low (L): 100–180 cm, medium (M): 200–340 cm, and high (H): 360–600 cm) at T1 and T2 [26]. In this study, sampling heights were divided into the same height ranges used by Chen et al. [26]. To test the variation in the results at different sampling height ranges, one-way ANOVA was performed between three height ranges (Table 2). The results showed that the differences (*p* < 0.05) of T0 and T1 were significant, and the differences in the 2.5–10 μm and >10 μm size < fractions between T2 and T3 at different height ranges were also significant (*p* < 0.05); in addition, the differences were always significant between every two height ranges (Appendix A).

The concentrations of different size fractions (0.02–1 μm, 1–2.5 μm, 2.5–10 μm, and >10 μm) in each height range are shown in Figure 5. The concentrations of fine particles (0.02–1 μm and 1–2.5 μm) did not change significantly with the increasing sampling height range for all trees; nevertheless, decreasing concentrations of 2.5–10 μm size fraction particles and increasing concentrations of >10 µm size fraction particles were found with increasing sampling height ranges for T0–T3 but not for T4.

## 4. Discussion

### 4.1. Effect of Enclosed/Open Environment

The potential of urban forests to reduce PM from the atmosphere and improve the quality of the urban environment has been confirmed in numerous studies. However, there is a difference in the PM retention capacity between trees located in the dense closed urban forest and trees in the open environment. There were significant differences in the weight of PM deposited on leaves, the SIRM, and the particle grain-size distribution between trees on the campus of Lanzhou University and those in the open road space. As shown in Table 1, the SIRM and weight of the PM of T0 were significantly higher than those of T1, T2, T3, and T4. In dense urban forests, the accumulation of PM on the leaf surface is high due to poor ventilation, long-term accumulation of PM, and limited cleaning conditions. Those trees surrounded by the crowns of adjacent trees experience reduced air circulation [25] and thus retain more atmospheric particles. In comparison, lower weights of PM and SIRM were observed at T1–T4, which were located in an open road space, where leaves are often washed by humans or by natural precipitation. Wang et al. [57] confirmed that a considerable proportion of the accumulated PM (28–48% of PM), especially most of the large and coarse particles [57,58] and the smallest size fraction (21–30%) [58] on leaves are easily removed by rainfall events. Simultaneously, as mentioned before, trees located in open spaces experienced good ventilation, which lead to more PM being removed from leaves by winds [59,60], especially high-speed winds [57].

A higher percentage of fine particles (0.02–1 μm, 1–2.5 μm, and 2.5–10 μm) was found at sites T1 and T2, which were close to the main road, suggesting that more fine particles produced by traffic activities were captured by needles, especially particles of the <2.5 μm size fraction. Robert et al. [61,62] and Zhu et al. [63,64] confirmed that most of the fine PM emitted from vehicle exhaust is in the size range of <1 μm, and Sgrigna et al. [9] found that traffic-related particle size distributions averaged 2.6 μm.

The lowest percentage of the >10 μm size fraction was observed at T1, indicating that fewer coarse particles were accumulated at T1, whereas fine particles deposited on the surface of needles mainly came from traffic activities, and natural dust fall deposited on the surface of needles included more >10 μm size fraction particles.

According to the results of ANOVA, significant differences (*p* < 0.05) in the weight of PM, SIRM, and particles in different grain-size fractions (0.02–1 μm, 1–2.5 μm, 2.5–10 μm, and >10 μm) were found in the four directions of T1 and T2, which were mainly due to the large amount of pollutants generated by traffic activities and the fact that roadside trees reduced air circulation. Simultaneously, roadside trees act as pollutant filters that influence the process of diffusion and accumulation of road PM derived from traffic activities, which can be verified by the SIRM and weight of particles in the NE direction, where the highest SIRM and weight of particles was found at T1, and those values decreased from T1 to T4 (from 6 to 24 m). In conclusion, dense urban forests are effective collectors of urban PM, and more particles with fine grain-size fractions (0.02–1 μm, 1–2.5 μm, and 2.5–10 μm) were captured by roadside trees were that were easily affected by traffic emissions. The rows of roadside trees act as effective filters for traffic PM, especially for finer particles.

### 4.2. Effect of Distance to the Road

The accumulation of PM on the leaf surface showed obvious changes with distance from pollution sources. As shown in Figure 3, a decreasing PM weight and SIRM were found with increasing distance from the road except for the SW direction. In this case, compared with the leaf magnetism at 6 m from the road, significant reductions were observed in the SIRM of up to 19.82% and 35.63% at 10 m and 24 m from the road, respectively, which indicated a significant decrease in the concentration of magnetic particles with increasing distance from the road where traffic activities (emission and disturbance) are the main source of magnetic pollution. Similarly, Szönyi et al. noted that the distance from the source (i.e., the vehicular traffic) is a crucial factor that increases the concentration-dependent magnetic parameters, ranging from the highest values closest to congested roads with stop and go traffic to low values within several meters [65]. Moreno et al. [38] found a significant decrease in susceptibility (45 to 9 × 10^−8^ m^3^/kg moving from 2 to 25 m of distance to the roadside) and IRM at a 25 m distance from the roadside. The weight of PM decreased 20.21% and 29.13% as the distance increased to 10 m and 20 m, respectively, which is consistent with the results near a large park, El Retiro, in Madrid (Spain) found by Gomez-Moreno et al. [66], with reductions in PM_10_ and PM_2.5_ of up to 25% at 20 m from the street both in summer and winter surveys, which indicated a positive impact from the vegetation via the reduction in traffic-induced PM_10_ and PM_2.5_ concentrations. In this study, the 0.02–1 μm size fraction decreased 6.02% and 5.36% at distances of 10 m and 24 m from the road, respectively, and the 1–2.5 μm size fraction decreased 4.46% and 2.55% at different distances, respectively. However, no significant reduction was observed in the 2.5–10 μm and >10 μm size fractions. This characteristic of variation was opposite to that reported by Gomez-Moreno et al. [66], who confirmed that PM_10_ and PM_2.5_ fractions were more easily removed by inertia mechanisms when tree canopies experienced wind, producing a stronger decrease than that seen for the fraction of PM_1_ in the largest park in Madrid with dense plants. This could be caused by the differences between the closed park and open space. In the closed park, PM is removed and diluted by dense trees via accumulation or absorption [67]. However, fine particles derived from traffic activities accumulate more easily than other particles on the *Juniperus formosana* needles, as mentioned in Section 3.1 and Section 4.1 (Figure 2).

Obvious differences in the PM weight and SIRM were observed with increasing distance between the four directions. The decreases were the largest in the NE direction both for the SIRM and PM weight, which were mainly caused by the important role of roadside trees, which removed a portion of traffic pollutants, especially those trees in the NE direction that directly faced traffic pollution sources and were on the windward side of the prevailing wind [26]. Moreover, it is worth noting that the PM weight and SIRM did not decrease but increased slightly in the SW direction, which could have been caused by the presence of a small concrete square, where residents take part in recreational activities that lead to dust resuspension and accumulation on the surface of needles, so that a slightly higher PM weight and SIRM were measured. In conclusion, roadside trees play an important role in the process of accumulation and sink of PM; with increasing distance from the road, decreasing PM was deposited on the needles.

### 4.3. Effect of Height

The differences in PM weight and SIRM between the three height ranges were significant for T0 and T1 (*p* < 0.0001) but not for T2, T3, and T4 (*p* = 0.109–0.999), indicating that the deposition of PM that might be caused by anthropogenic activities near the ground on the needles of trees in closed urban forests and rows of roadside trees is affected by sampling height. Nonsignificant differences in the 0.02–1 μm and 1–2.5 μm size fractions were found between different height ranges (Appendix A), which may have been caused by the penetrability of fine particles through the entire sampling height of the tree crowns. However, significant differences (*p* < 0.0001 and 0.001–0.047, respectively) were found between the L and H height ranges and the L and M height ranges for T1, T2, and T3, and significant differences (*p* = 0.004, 0.023) were also found between L and H for T0. Under the influence of human activities near the ground, a number of fine particles are resuspended, which increases the percentage of fine particles deposited on the leaves. Simultaneously, as mentioned above, due to the penetrability of the finer particles, the distributions of the 0.02–1 μm and 1–2.5 μm size fractions were homogeneous throughout the entire tree crown, and decreasing concentrations of the 2.5–10 μm size fractions and increasing concentrations of the >10 μm size fractions were found with increasing height due to reduced human activities and the deposition of natural dust on the tops of tree crowns.

## 5. Conclusions

The analysis of magnetic variability and grain-size distribution of particles deposited on leaves at different distances from the road showed the impact of pollution sources on the leaf magnetic properties and particle deposition. The results showed that the trees located in dense urban forests retained more PM than trees located in an open space. In addition, the SIRM and PM weight decreased with increasing distance to the road, suggesting that distance from traffic pollution sources plays a key role in reducing the air concentration of PM. The grain-size frequency distributions of the particles deposited on the surfaces of all leaves showed a unimodal spectrum on a frequency curve with a main peak of approximately 20 μm and a minor peak between 0.1 and 1 μm, indicating that the particles deposited on the *Juniperus formosana* needles were mainly fine particles. Urban forests play an important role in the removal of PM, and the characteristics of PM (SIRM, weight, and grain size) deposited on leaves change with the distance of trees from the road, wind direction, and the planting combination of trees.

## Figures and Tables

**Figure 1 ijerph-18-11964-f001:**
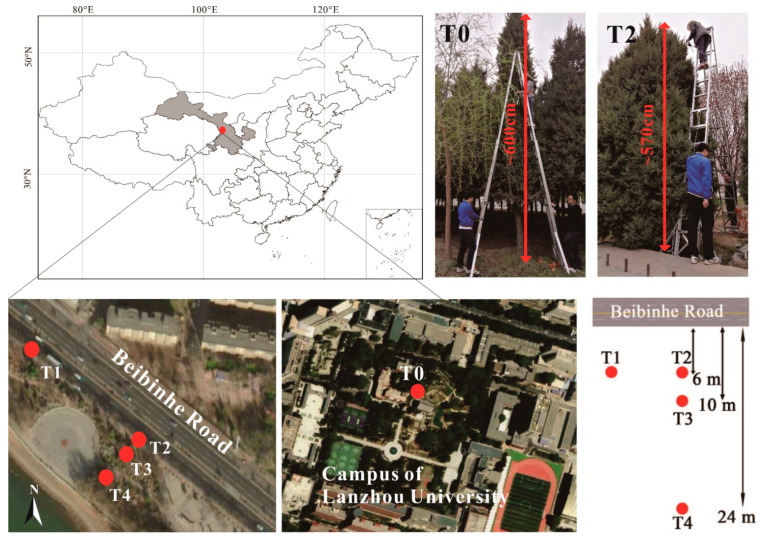
Sampling location in Lanzhou.

**Figure 2 ijerph-18-11964-f002:**
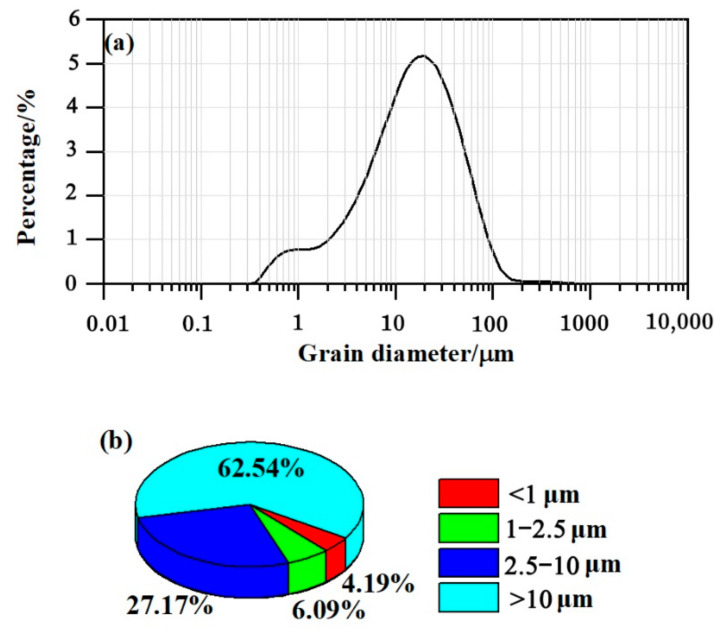
(**a**) The particle size frequency distribution curve of particles deposited on needles. (**b**) The percentages of the particle size fractions (0.02–1 μm, 1–2.5 μm, 2.5–10 μm, and >10 μm).

**Figure 3 ijerph-18-11964-f003:**
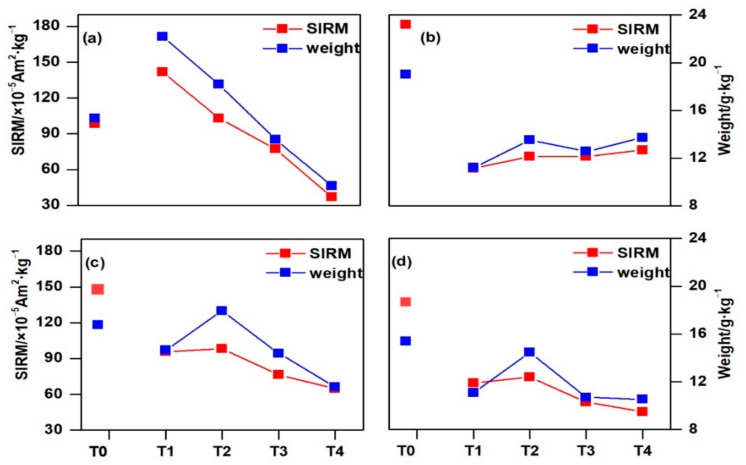
Variation in the SIRM and weight of PM deposited on the needles at different sampling sites (T0–T4) in four directions ((**a**)—NE, (**b**)—SW, (**c**)—NW, (**d**)—SE)).

**Figure 4 ijerph-18-11964-f004:**
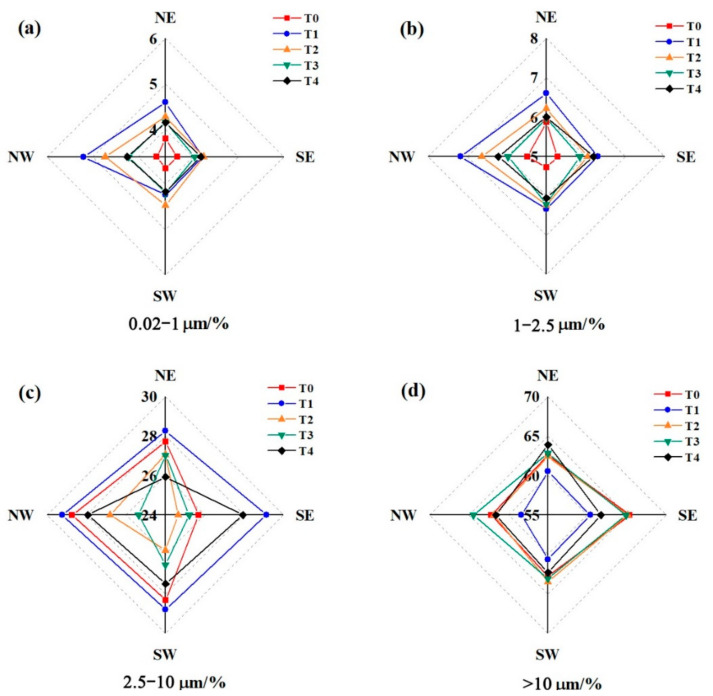
Directional percentages of (**a**) 0.02–1 μm, (**b**) 1–2.5 μm, (**c**) 2.5–10 μm, and (**d**) >10 μm particles deposited on needles at sites T0–T4.

**Figure 5 ijerph-18-11964-f005:**
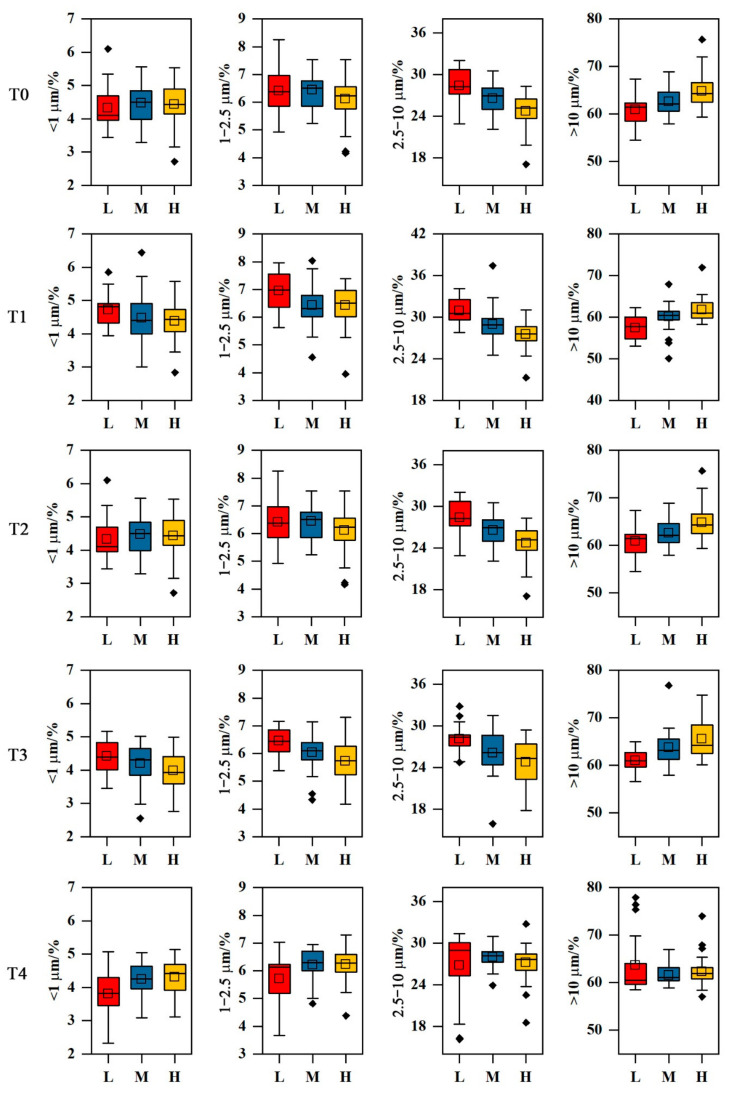
Box and whisker plots showing the percentage of different size fractions at different sampling height ranges (L, M, and H) for T0–T4.

**Table 1 ijerph-18-11964-t001:** Results (SIRM, weight of PM, and different grain-size particles (0.02–1 μm, 1–2.5 μm, 2.5–10 μm, and >10 μm)) of one-way ANOVA performed between four sampling directions (NE, SW, NW, and SE) at sites T0–T4. Statistically significant differences (*p* < 0.05) are highlighted in bold.

	T0		T1		T2		T3		T4	
	F	*p*-Value	F	*p*-Value	F	*p*-Value	F	*p*-Value	F	*p*-Value
SIRM	3.795	0.013	17.821	<0.0001	10.785	<0.0001	5.454	0.002	11.540	<0.0001
Weight of PM	1.040	0.378	10.213	<0.0001	5.269	0.002	2.067	0.110	3.714	0.014
0.02–1 μm	0.545	0.653	12.178	<0.0001	3.680	0.015	0.611	0.610	0.126	0.944
1–2.5 μm	3.862	0.012	8.254	<0.0001	2.965	0.036	1.318	0.274	0.584	0.627
2.5–10 μm	4.832	0.004	1.373	0.256	4.609	0.005	2.026	0.116	2.745	0.048
>10 μm	4.305	0.007	4.705	0.004	4.206	0.008	1.823	0.149	1.858	0.143

**Table 2 ijerph-18-11964-t002:** Results (SIRM, PM weight, and different grain-size particles (0.02–1 μm, 1–2.5 μm, 2.5–10 μm, and >10 μm)) of one-way ANOVA performed between three height ranges (L, M, and H) at sites T0–T4. Statistically significant differences (*p* < 0.05) are highlighted in bold.

	T0		T1		T2		T3		T4	
	F	*p*-Value	F	*p*-Value	F	*p*-Value	F	*p*-Value	F	*p*-Value
SIRM	13.349	<0.0001	14.142	<0.0001	0.054	0.948	3.364	0.039	1.935	0.150
PM weight	31.777	<0.0001	27.794	<0.0001	0.288	0.751	1.218	0.300	4.509	0.014
0.02–1 μm	7.185	0.003	1.588	0.210	0.411	0.664	4.334	0.016	6.137	0.003
1–2.5 μm	3.613	0.031	3.218	0.045	2.388	0.097	8.811	<0.0001	4.712	0.011
2.5–10 μm	4.852	0.010	19.351	<0.0001	16.465	<0.0001	8.274	<0.001	1.566	0.215
>10 μm	6.941	0.002	14.186	<0.0001	11.456	<0.0001	11.844	<0.0001	1.615	0.205

## Data Availability

All relevant data sets in this study are described in the manuscript.

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
