# Peer review of "Magnetism and Grain-Size Distribution of Particles Deposited on the Surface of Urban Trees in Lanzhou City, Northwestern China"

_ijerph, 2021, doi:10.3390/ijerph182211964_

Round 1
Reviewer 1 Report
The paper is scientifically good. Below, I provide, not only, but mostly suggestions for authors:
1 - Line 54: When refering the tree species (Juniperus formosana) studied, authors can briefly refer which other works studied same species, similar tree species and other different trees/folieage/canopies. Also a photo of all/some of the trees, and/or some discussion revealing the healthy state of them could be provided.
2 - Do check Line 110 about range of measuements;
3 - figure 2 is subdivided into a) and b) but not the legend;
4 -In line 144 is better to include "x10-5";
5 - line 233 trees is mispeled;
6 Setence in lines 306-308 is not clear. consider revising it; Not sure if the diffussion is the better word. Sink? The dispersion around and inside the tree (as a porous meddium) is not evaluated here. It can mislead readers.
7 - conclusion in lines 331-333 is very plausible but not measured in the paper and thus should be rewritten as an hypothesis that can be supported also with data here measured. I mean that the "ventilation" was not measured in this work. However I do agree that it one obvious strong explanations. This can also be improved adding a supporting reference here.
8. Similarly the observation explained in line 334 could be explained by the air concentration of PM that decreases with increase from the road source. Again this should be presented as a very likely factor as PM air levels were not registered.
9 - Line 340: "different environmental conditions" could be better explained with examples of those conditions. Distante between trees and the road source? Wind velocity? Tree type and nearby obstacles?
Reviewer 2 Report
This article is a concise contribution about magnetic biomonitoring methods applied in Lanzhou, China.
The manuscript is well written and deserves to be published after moderate revisions, which should be addressed to a more accurate description of the magnetic methods and some other aspects that should be discussed or at least introduced in the text.
1) the magnetic methods should be written more in detail: it is reported only a reference, which is not enough for a standalone paper.
2) PM was weighed after washing, but how can you exclude that a part of the magnetic particles remained immobilized, or accumulated, inside the leaves, and was not removed by washing, especially for the finest fraction? Can you exclude that a part of the PM fraction is soluble? There are many studies about the partial efficiency of washing leaves for removing the particles, depending on trichomes, waxes etc...
3) Which are the sources of airborne particulate matter accumulated by the T0 tree? Do you think that they arise from the roadside? Or can you suppose an influence from the campus activities, considering that it is strange to observe an higher concentration of magnetic particles at T0, after demonstrating that the distance from the roadside is reduces SIRM. Is the use of a single tree (T0) statistically significant?
4) Can you infer something about the relationship between sources and grainsize? Do you think the coarser particles are anthropic or natural?
5) In the following papers, there are several aspects about PM accumulation in leaves and the relationship between magnetism and distance from the roadside. Muhammad et al. made other articles on the same subject.
Muhammad, S., Wuyts, K., Samson, R., (2019). Atmospheric net particle accumulation on 96 plant species with contrasting morphological and anatomical leaf characteristics in a common garden experiment. Atmospheric Environment 202: 328 – 344.
Szönyi, M., Sagnotti, L., Hirt, A. M., A refined biomonitoring study of airborne particulate matter pollution in Rome, with magnetic measurements on Quercus Ilex tree leaves, Geophysical Journal International, Volume 173, Issue 1, April 2008, Pages 127–141, https://doi.org/10.1111/j.1365-246X.2008.03715.x
With my best regards to the Editor and the Authors
Reviewer 3 Report
Work done in a clear and thoughtful way. It raises very important issues in the field. Developed results well described. Literature cited very well. I have no objections to the current form of presentation. I believe it can be published in its current form
Round 2
Reviewer 2 Report
Dear Editor and authors,
thank you for taking into account my review, even if some points were not completely solved, and the revision process was not particularly in deep.
Anyway, I think that many aspects of this paper can be interesting and deserve publication, and for this I suggest minor revisions, taking great care of the English form. For this, I attach an annotated pdf file, with suggestions for improving the syntax.
With regards,
the reviewer
